# Vision-Based Assessment of Skeletal Muscle Decline: Correlating Gait Variance with SPPB Performance

**DOI:** 10.3390/healthcare13121405

**Published:** 2025-06-12

**Authors:** Zhaozhen Tong, Sinan Chen, Yuko Yamaguchi, Masahide Nakamura, Hsin-Yen Yen, Shu-Chun Lee

**Affiliations:** 1Graduate School of Economics, Faculty of Economics, Kobe University, 2-1 Rokkodai-cho, Nada-ku, Kobe 657-8501, Japan; 2Center of Mathematical and Data Sciences, Kobe University, 1-1 Rokkodai-cho, Nada-ku, Kobe 657-8501, Japan; masa-n@cs.kobe-u.ac.jp; 3Graduate School of Health Sciences, Kobe University, 7-10-2 Tomogaoka, Suma-ku, Kobe 654-0142, Japan; y.yuko@port.kobe-u.ac.jp; 4School of Gerontology and Long-Term Care, Taipei Medical University, No. 250, Wuxing St., Taipei 11031, Taiwan; kenji@tmu.edu.tw (H.-Y.Y.); sclee@tmu.edu.tw (S.-C.L.)

**Keywords:** elderly fall risk, gait analysis computer, vision, short physical performance battery, non-invasive assessment

## Abstract

Background: With the global population aging, the proportion of the elderly is increasing, leading to health challenges. The decline in the elderly’s physical function raises their fall risk, which affects their health and burdens the healthcare system. Traditional fall risk assessment methods like Short Physical Performance Battery (SPPB) have limitations, while computer vision technology shows potential but also has drawbacks. Objective: This study aims to use computer vision technology to quantify the elderly’s gait movement features, analyze their correlations with SPPB test scores and duration consumption, and explore a solution for long-term monitoring and more efficient fall risk assessment. Methods: Data from 19 elderly Japanese subjects, including SPPB test data and camera-captured body movement data, were analyzed. Python (Version 3.12.6) was used to obtain JSON data, calculate movement distances, and construct a comprehensive index. Correlation analysis and principal component analysis (PCA) were performed. Results: The variance mean indicator of the comprehensive index associated with movement distance had a significant negative correlation with the completion duration of Test 2 in SPPB, indicating that greater gait variability might be related to better physical vitality. PC1 (Muscle-Control Reserve) and PC2 (Learning-Fatigue Response) obtained from PCA had a positive relationship with the test duration. The comprehensive index had a positive but not highly significant correlation with test scores. Conclusions: This study analyzed the correlation between the elderly’s gait movement features and SPPB test performance. It innovated in data collection and analysis methods. Future research can be improved by expanding the sample size, adding more parameters, and applying deep-learning techniques.

## 1. Introduction

The global population is aging at an unprecedented rate, with the proportion of elderly individuals projected to increase significantly in the coming decades. This demographic shift brings with it a host of health-related challenges. Among them, the decline in the physical function of the elderly poses a serious challenge to their physical health. Specifically, the deterioration of muscle strength and balance in the elderly not only diminishes their quality of life but also poses a significant risk of falls, leading to severe injuries and a substantial burden on healthcare systems [1]. Early detection and intervention are thus of paramount importance to mitigate these risks and ensure the well-being of the elderly population [2].

Traditional methods of assessing fall risks in the elderly often rely on clinical tests such as the SPPB, research shows that the total score of SPPB and the balance ability are directly correlated with fall incidents, and the assessment efficacy of this test is comparable to that of the commonly used Performance Oriented Mobility Assessment (POMA) tool [3]. However, such tests are time-consuming and require specialized personnel to administer. Moreover, these methods do not provide continuous monitoring, which is essential for timely intervention [4]. In recent years, the advent of computer vision technology has opened up new possibilities for continuous health monitoring. This study proposes a novel approach that leverages computer vision to quantify gait movement features as a means of early warning for long-term monitoring of fall risks in the elderly [5]. By analyzing the correlations between these gait features and the scores and durations of the SPPB test, we aim to provide a more efficient and accessible solution to this pressing issue. Similar computer vision approaches have been successfully applied to motor assessment in Parkinson’s disease and gait analysis using single RGB cameras [6,7].

To clearly illustrate the technical framework of this study, a conceptual diagram (Figure 1) is presented, outlining three core stages—data collection, feature analysis, and verification—to visually demonstrate the research process. In data collection, a single camera is used to capture the coordinate data of skeletal key points during the SPPB test. For feature analysis, first, the Euclidean distances of gait movement, along with their variances and mean values, are used to feature gait fluctuations. Then, *Principal Component Analysis (PCA)* is employed for dimensionality reduction to extract key features that can comprehensively reflect an individual’s gait pattern. In the verification stage, the extracted gait features are correlated with the durations and scores of the SPPB test. By analyzing the correlation strength and statistical significance, the effectiveness of these features as indicators for fall risk assessment is verified.

In this study, a single camera was employed to collect the skeletal key point data of the subjects. There is no need for the subjects to wear identification devices or complex sensing instruments. The gait movement data can be accurately extracted through visual algorithms, making this method applicable to scenarios such as communities and families. The analysis focuses on the spatiotemporal fluctuation features of the gait movement distance. Since it integrates the two-dimensional information of distance and fluctuation, it can reflect an individual’s gait pattern more comprehensively and meticulously. At the same duration, it is closely related to the fall risk. For example, among 50 studies, 29 clearly pointed out that the walking speed of those who fell was significantly lower than that of those who did not fall; 19 out of 39 studies showed that the step length of the fallers was shorter; among 18 studies, 9 found that the double support phase of the fallers was longer and its variability was also higher. However, most of the existing studies mainly revolve around gait spatiotemporal parameters such as walking speed, step length, and double support phase, while systematic research on the features of gait movement distance and fluctuation remains relatively scarce. Therefore, the analysis of this feature in this study is not only expected to deepen the understanding of the relationship between gait patterns and falls, but also provide a new perspective and potential indicators for fall risk assessment [8].

This approach builds on established techniques for real-duration human pose estimation from single RGB cameras and recent advancements in reliable markerless systems for gait monitoring [9,10]. By calculating the Euclidean distances of the left and right feet movement, variances, and the mean of variances during the two repeated tests of the subjects in Test 2 (walking test), a comprehensive index is constructed. Additionally, *PCA* is employed to verify the effectiveness of this index in representing individual differences. It is worth noting that this method using a single-camera setup offers a low-cost and contactless solution for evaluating gait in the elderly, and the PCA-derived comprehensive index can effectively capture individual gait differences, providing a basis for personalized fall risk assessment.

Subsequently, in the Section 2, we will compare our study with previous research in aspects like equipment usage, privacy protection, and camera capture methods to clarify the advantages of our approach [11]. The Section 3 and Section 4 detail the data collection from 19 elderly subjects, the process of constructing the comprehensive index [12]. In the Section 5, we will present and analyze the correlations between the comprehensive index and SPPB test scores and durations. Finally, the Section 6 will summarize the research, emphasize our study’s innovations, and suggest future research directions.

The research findings indicate that there is a significant correlation between this index and the test scores as well as the action durations, suggesting that the fluctuation pattern of the gait movement distance can objectively reflect the execution quality of the sit-to-stand action [13]. This method can achieve contactless assessment of motor function merely by using single-camera visual sensing devices, providing a quantifiable and easily popularized technical pathway for the long-term monitoring of fall risks among the elderly in communities [14]. Meanwhile, it lays a data foundation for the formulation of personalized fall risk intervention strategies. By offering a non-invasive and continuous monitoring solution, this study aims to contribute to the development of more effective and accessible healthcare services for the elderly, ultimately improving their safety and quality of life.

## 2. Related Work

This section first reviews traditional fall risk assessment methods for the elderly (such as SPPB), indicating their relationship with fall risks and limitations. It then summarizes the latest applications of computer vision technology in fall risk assessment and gait analysis, discussing the advantages and disadvantages of existing studies in terms of equipment complexity, privacy protection, and data collection scenarios. Finally, a comparative analysis of methods and equipment across different studies is conducted, emphasizing the design of the single-camera scheme adopted in this study in terms of operational simplicity and privacy protection, as well as its features in data stability and adaptability to diverse environments.

### 2.1. Preliminaries

In the context of the accelerating global aging process, the health management of the elderly faces numerous challenges. Among them, the assessment of fall risks is particularly crucial. Traditional assessment methods, such as the Short Physical Performance Battery (SPPB), can evaluate the physical function of the elderly to some extent. However, the testing process is time-consuming and labor-intensive, relying on professional personnel for operation [15]. Moreover, it cannot achieve continuous monitoring of the daily activities of the elderly, making it difficult to meet practical needs [16]. In recent years, computer vision technology, with its unique advantages, has been widely studied and applied in the field of healthcare, especially in the assessment of fall risks and gait analysis for the elderly [17,18].

As show in Table 1. studies by Sarapata et al. [6], Ziyang Wang et al. [7], Cedeno-Moreno et al. [19], Boldo et al. [20], and Lim et al. [21] have explored the application of computer vision in this field from different perspectives. Some of them used specific device combinations for gait analysis or disease-related motor assessments, some verified the reliability of relevant systems, and others conducted fall risk predictions. However, these studies have their own advantages and disadvantages in terms of equipment usage, privacy protection, and data collection [22,23]. For example, some equipment is complex and expensive, there are potential risks in privacy protection, and data acquisition is restricted.

Based on the above-mentioned research achievements and shortcomings, this study is of great significance for exploration [24]. This study plans to use a single-camera combined with computer vision technology to quantify the gait fluctuation features reflected by the skeletal key points of the elderly during walking, and deeply explore the relationship between these features and the results of the SPPB test. On the one hand, the use of a single camera is expected to simplify the equipment configuration, reduce costs, and improve the convenience of monitoring, making it more suitable for promotion and application in community and home environments [25]. On the other hand, focusing on gait fluctuation features can more accurately evaluate the physical function status of the elderly from the perspective of dynamic changes, providing a more effective basis for fall risk assessment [21]. This not only helps to fill the gaps in existing research but also may provide innovative solutions for the early warning and personalized intervention of fall risks in the elderly, promoting the development of the elderly health management field.

### 2.2. Comparison of Equipment and Methods

A comparison between this study and the research by Sarapata et al. [6], Wang et al. [7], Cedeno-Moreno et al. [19], Boldo et al. [20], and Lim et al. [21] clearly reveals the differences and advantages/disadvantages in aspects such as equipment usage, privacy protection, and camera capture methods.

In terms of equipment usage, this study has a distinct advantage as it only uses one fixed camera. Sarapata et al. [6] used consumer-grade handheld devices in conjunction with OpenPose. Although handheld devices are common, OpenPose, as an auxiliary pose estimation device, adds complexity to the system. In the study by Ziyang Wang et al. [7], they used a Double Robot, an iPad, and an Intel RealSense D415. The combination of these devices is complex and costly. Moreover, the scenarios for the collaborative work of the robot and multiple devices are relatively limited, restricting flexibility to some extent. Cedeno-Moreno et al. [19] used one IDS UI-3130CP-M-GL R2 monochromatic camera and one electric treadmill. The use of a treadmill increases the equipment cost and site requirements, and the application scenarios are greatly restricted. Lim et al. [21] used two fixed cameras and relied on the inertial sensor data from the Mendeley public dataset. Although the number of cameras is not large, the dependence on the public dataset limits the acquisition and use of data, and obtaining additional sensor data may increase the complexity of the research. In contrast, the equipment used in this study is simple, low-cost, easy to install and deploy, and more conducive to quickly setting up a research environment in various scenarios [19,26].

Regarding privacy protection, in this study, only human key-point data are captured through the camera during data collection, without recording actual video footage, ensuring the privacy of subjects from the source. However, most of the other studies first obtain video data and then extract stick-figure-like data by processing the video. For example, although Sarapata et al. [6], Boldo et al. [20], and Cedeno-Moreno et al. [19] obtained informed consent from the subjects and followed the approvals of the ethics committees, and used OpenPose or other methods to extract human joint key-point data, respectively, there are still privacy risks in the process of extracting data from videos. They may face challenges in privacy protection during the actual detection process. Lim et al. [21] obtained approval from the ethics committee, informed consent, and anonymized the data, and used AlphaPose to extract human key-point data. However, there are still potential privacy risks in the process of video data processing. Lyu et al. [27] did not clearly mention whether they considered privacy and specific measures, nor did they clarify whether they only used stick-figure-like data. There may be the use of original video data, and the privacy protection situation is uncertain, which may pose a threat to the privacy of subjects in practical applications. Thus, it can be seen that the privacy protection measures in this study are more direct and effective, reducing risks from the source of data collection [28,29].

In terms of the camera capture method, this study adopts the static fixed-camera recording method. This method facilitates stable data collection, can reduce external interference, and ensure the consistency and accuracy of data. Sarapata et al. [6], Boldo et al. [20], Cedeno-Moreno et al. [19], and Lim et al. [21] also adopted static capture methods. Although these methods can ensure the stability of data collection, they lack flexibility when capturing movements in complex scenarios. For example, Cedeno-Moreno et al. [19] fixed the camera to shoot treadmill walking, which is limited to the treadmill scenario, and the universality of the data may be affected. Lim et al. [21] fixed the camera to shoot the TUG test, and the application scenario is relatively single. Lyu et al. [27] adopted dynamic capture, that is, the robot moves and follows the subject for shooting. This method can flexibly track the shooting object, but it has extremely high requirements for the stability and accuracy of the equipment, and the shooting process is easily affected by environmental factors, resulting in poor data stability. In comparison, the static fixed-camera recording method in this study can ensure data quality and can be applied to various scenarios by optimizing the shooting plan, showing better adaptability.

## 3. Methodology

This section introduces the research methodology and describes the data-related content. The study included 19 elderly Japanese subjects, with basic data covering age, gender, height, and weight distributions. Analysis of test scores and durations showed that the first test had a greater impact on the total score but required longer duration; the second test had higher scores and shorter duration, which can more accurately reflect the true physical function. Analysis of movement distance patterns indicated significant individual differences in stride length change trends among subjects across the two tests, reflecting the complexity of the association between movement patterns and physical function.

### 3.1. Data Description

#### 3.1.1. Baseline Information

This study included 19 elderly Japanese subjects from a certain area. The protocol was approved by the Institutional Review Board (IRB) of Kobe University Graduate School of Health Sciences (No. 1249). The baseline data were distributed as follows, as shown in Figure 2:Age: 60–80 years, 9 subjects, 47.4%; over 80 years, 10 subjects, 52.6%.Gender: 15 males, 78.9%; 4 females, 21.1%.Height: 145–155 cm, 5 subjects, 26.3%; 155–165 cm, 5 subjects, 26.3%; 165–175 cm, 8 subjects, 42.1%; 175 cm or above, 1 participant, 5.3%.Weight: 40–50 kg, 4 subjects, 21.1%; 50–60 kg, 5 subjects, 26.3%; 60–70 kg, 9 subjects, 47.4%; 85 kg or above, 1 participant, 5.3%. It is noteworthy that there were no subjects within the 70–85 kg range.
Figure 2Basic information of subjects.
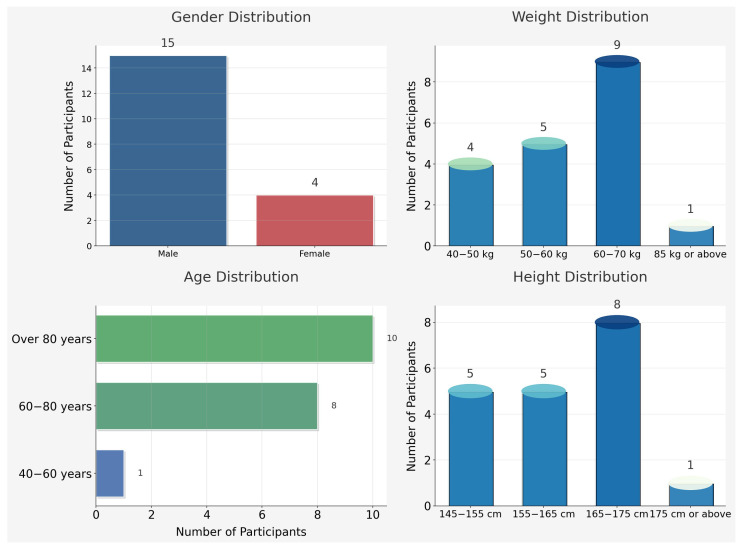



#### 3.1.2. Score Distribution and Test Duration

Figure 3 shows that the distribution of the first test scores closely aligns with the total scores, indicating that the first test significantly contributes to the overall score [30]. In contrast, the second test scores are generally higher than the total scores, suggesting that the second test has a relatively smaller impact and may not fully reflect the subjects’ actual physical function. Additionally, an analysis of test duration reveals that the first test takes significantly longer than the second, indicating that total scores are largely influenced by the first test’s duration. However, the extended duration of the first test might be due to an adaptation process rather than an accurate reflection of physical ability. Therefore, the shorter duration and higher scores in the second test could provide a more precise representation of subjects’ true physical condition [31].

#### 3.1.3. Movement Distance Distribution

As shown in Figure 4, subjects exhibited considerable individual differences in movement patterns across the two tests, with no clear overall trend [32]. Some showed a gradual increase or decrease in stride length during the test, while others shifted from consistent small steps to larger strides, or vice versa. This variability in movement patterns is consistent with findings from video-based activity recognition studies, which also observed significant individual differences in movement execution. These variations may reflect differences in adaptation to the experiment and diverse movement behaviors among individuals [6,33]. This highlights the complex relationship between movement patterns and physical function, providing valuable insights for refining assessment tools.

## 4. Proposed Method

This section details a four-step data analysis methodology developed to analyze movement data and their correlation with physical function scores. First, Python (Version 3.12.6) is used to extract foot position coordinates from camera-captured JSON data, with Euclidean distance algorithms applied to calculate movement distances during the SPPB Test 2. Second, variance metrics of movement distances are computed to construct comprehensive indicators that capture individual variability. Third, *PCA* is employed to reduce data dimensionality to two principal components (PC1 and PC2), with a cumulative variance contribution rate of 100% to ensure complete retention of data information. Finally, correlation analysis and heatmaps are used to identify associations between comprehensive movement distance indicators and Test 2 scores/durations, aiming to establish quantitative links between gait patterns and physical function assessments.

### 4.1. Step 1: Pedestrian Movement Data Extraction and Analysis

Python is used to obtain the JSON data captured by the video and extract the coordinates of the foot positions based on the data structure, similar to the approach used by Wang et al. [7] for single RGB camera gait analysis [10,34]. The skeletal key point tracking methodology is inspired by real-duration pose estimation techniques [10]. By applying the Euclidean distance algorithm Equation (Equation 1), we calculated the movement of the left and right feet of the subjects in the SPPB experiment during the Test 2 and observed the distribution. Figure 4 shows the distribution diagram.(1)$$d_{k(x_{i})}=\sqrt{{\left(x_{\left(k+1\right)\left(x_{i}\right)}-x_{k(x_{i})}\right)}^{2}+{\left(y_{\left(k+1\right)\left(x_{i}\right)}-y_{k(x_{i})}\right)}^{2}}$$(2)$$\mathrm{Var}\left(d_{\left(x_{i}\right)}\right)=\frac{1}{N-1}\sum_{k=1}^{N-1}{\left(d_{k(x_{i})}-\bar{d_{\left(x_{i}\right)}}\right)}^{2}$$(3)$$\mathrm{var}\text{\_}\mathrm{mean}=\frac{\sum_{i=15}^{16}\left(\mathrm{Var}{\left(d_{\left(x_{i}\right)}\right)}_{\mathrm{test}2\text{\_}1}+\mathrm{Var}{\left(d_{\left(x_{i}\right)}\right)}_{\mathrm{test}2\text{\_}2}\right)}{4}$$

Equation (Equation 2) is used to calculate the variance in gait movement distance. By dividing by the count of movement distance data minus one, the result can reflect the unbiased estimation of the fluctuation magnitude of motion distance data. Equation (Equation 2) is employed to compute the mean value of gait movement distance variances. It selects the data corresponding to the positions of the left and right feet, sums this variances under two different test conditions, respectively, aggregates these sums, and then divides by four to obtain the average level of this variances in two tests of Test 2, thereby assisting in evaluating the overall trend of motion distance fluctuations. In the formulas, *N* represents the number of gait movement distances within the test duration period and *i* represents the positions of foot positions, where i=15 and i=16 denote the left and right feet.

### 4.2. Step 2: Calculate the Variance of Movement Distance and Construct a Comprehensive Indicator

To better reflect the changes in movement distance of subjects during their motion, as shown in Figure 5, we calculated the variance in gait movement distance and the mean of the variance.

As can be seen from the distribution diagram, in the Test 2 experiment, the variance in the left and right foot movement distances (var15_test_2.1, var15_test_2.2, var16_test_2.1, var16_test_2.2) in the two consecutive experiments varies among individuals [35]. Some individuals have similar variances in the left and right foot movement distances between the two experiments, while others have a large difference in the variances in the left and right foot movement distances between the two experiments [9].

The variance mean of a single test (var_mean_2_1, var_mean_2_2) represents the average variance in left and right foot movement distances within a single test, as calculated by the method described in Equation (Equation 2). This calculation approach only captures the local stability of both feet during a single test, without integrating information across multiple tests. As a result, for different individuals, this value often closely resembles the variance data of a single foot, indicating instability [36].

Conversely, var_mean is computed by aggregating the variance data of both feet from two tests, following the logic outlined in Equation (Equation 3). By comprehensively averaging the variance information from both feet across two tests, var_mean consistently falls in the middle range among different individuals and effectively reflects inter-individual differences. Thus, it serves as a more reliable comprehensive indicator [37].

### 4.3. Step 3: Construct a Comprehensive Indicator Through Principal Component Analysis (PCA)

Through the method of *PCA*, it can be observed that the variance contribution rates of PC1 and PC2 are approximately 0.657 and 0.343, respectively, summing up to 100% [38]. This allows for a significant retention of data information. Therefore, it is decided to reduce the dimensionality of the movement distance data to two dimensions [11].

### 4.4. Step 4: Correlation Analysis

By calculating the correlation coefficients between the comprehensive indicator of movement distance and the scores and duration taken by subjects in Test 2, we can identify any correlations [39]. Additionally, heatmaps are used to visualize the correlation relationships [32].

## 5. Results and Discussion

This section presents the analysis results, which show a significant negative correlation between the mean of walking distance variance (Mean) and SPPB test completion durations (duration, duration1, duration2), suggesting that greater gait variability may be associated with better physical vitality or coordination. Although not statistically significant, duration variables exhibited positive trends with principal components PC1 (muscle-control reserve) and PC2 (learning-fatigue response). Combined with the analysis of gait test duration distributions, this indicates that muscle control ability and fatigue accumulation may influence test duration. Regarding SPPB test scores, while correlations with comprehensive indicators did not reach significance, consistent positive trends across different tests were observed, hinting at potential associations worthy of further exploration. These findings contribute to understanding the relationship between gait features and physical function in older adults, providing a basis for computer vision-based fall risk assessment.

### 5.1. Purpose and Overview

This study aims to utilize computer vision technology to assess the fall risks of the elderly by quantifying their gait movement features and to explore the relationships between these features and the performance on the SPPB [19]. The research collected the skeletal key-point data of 19 elderly subjects, and constructed a comprehensive index and *PCA* [40]. Subsequently, a correlation analysis was conducted between the comprehensive index and the scores and durations of the SPPB test.

In this section, the research results will be elaborated in detail. It is crucial to acknowledge that with a sample size of only 19 subjects, the study has certain limitations. The results may predominantly reflect the gait features of these specific individuals, and thus, the generalizability of the findings is restricted. However, it should be noted that in the preliminary exploration stage of this research field, a relatively small sample size is common. This is because it helps to quickly obtain some valuable information initially, which can guide subsequent larger-scale research. Moreover, based on the distribution of basic information, it can be demonstrated that the sample has a certain degree of representativeness. Despite these limitations, we will explore the relationships between the comprehensive index and the completion durations and scores of the SPPB test in depth [21]. By analyzing these results, it is possible to not only reveal the potential links between gait fluctuation features and the physical function of the elderly but also provide important evidence for the assessment of fall risks in the elderly based on computer vision technology [39]. At the same time, this study will be compared with previous relevant research to further clarify the value and significance of the research findings, laying a foundation for subsequent discussions and conclusions [41]. The ultimate goal is to promote the development of the field of fall risk assessment in the elderly and provide scientific support for the formulation of more effective intervention measures.

### 5.2. Result 1: The Relationship Between Comprehensive Indicators and Test 2 Duration in SPPB

In this study, the data were first classified into binary variables, ordinal variables, and continuous variables. Specifically, gender is a binary variable, test scores are ordinal variables, and other variables, such as body composition measurements, are continuous variables [1]. In the selection of correlation analysis methods, this study is strictly based on the features of variable types. Pearson’s correlation coefficient is suitable for measuring the linear relationship of continuous variables, as it can effectively capture the numerical change trends among variables. For ordinal variables, since their main feature is the ranking order, Kendall’s Tau correlation coefficient and Spearman’s rank correlation coefficient are more adept at reflecting the monotonic relationships between them, accurately capturing the associative changes in variable orders. When a binary variable is combined with a continuous variable, the point-biserial correlation coefficient can well quantify the degree of association between these two different types of variables. In the case of two binary variables, considering that traditional correlation concepts have limited applicability, a default value is used for processing. In this way, the scientificity and rigor of the correlation analysis in this study are ensured.

Based on this classification, appropriate correlation analysis methods were selected for different variable combinations: for two binary variables, a default value was returned; for combinations involving one binary variable, the point-biserial correlation coefficient was used; for two ordinal variables, Kendall’s Tau correlation coefficient was applied; for two continuous variables, Pearson’s correlation coefficient was calculated to measure linear relationships; and for mixed-type variable combinations, Spearman’s rank correlation coefficient was used to assess associations [2]. Prior to computation, ordinal variables were converted to ranks and standardized, while continuous variables were smoothed using Locally Weighted Scatterplot Smoothing (LOWESS) to reduce noise interference. To control for false positives due to multiple comparisons, p-values were adjusted using the False Discovery Rate (FDR) correction method, ensuring the accuracy and reliability of the correlation analysis [15].

The study found a certain negative correlation between the mean of walking distance variance (Mean) and SPPB test completion durations (duration, duration1, duration2), indicating that greater gait variability is associated with shorter test durations. Specifically, mean shows negative correlations with duration (−0.87***), duration1 (−0.86***), and duration2 (−0.96***).

Additionally, according to the results of the principal component analysis in Figure 6, the variance weights of the walking movement distances in the two tests of Test 2 corresponding to pc1 are both 0.5. For pc2, the variance weight of the walking movement distance in the first test of Test 2 is 0.5, and the weight in the second test is −0.5. Combining with the results of the correlation analysis in Figure 7, the details are as follows:
**PC1** **(Muscle-Control Reserve)**: The four variables var15_test2.1, var16_test2.1, var15_test2.2, and var16_test2.2 all exhibit a positive loading of 0.50 on PC1. This indicates a pattern of synchronous changes in relevant gait indicators across different test phases and limb movements. From a biomechanical standpoint, when an individual has an adequate muscle reserve, it provides the material basis for flexible gait adjustments, leading to a more stable walking performance. Therefore, PC1 embodies the "basic support" role of muscles in gait regulation and reflects an individual’s potential to adjust gait as a whole based on muscle strength.The correlation between PC1 and duration is 0.24, suggesting a slight positive trend.**PC2** **(Learning-Fatigue Response)**: Statistically, variables in Test 2.1 phase (var15_test2.1, var16_test2.1) exhibit a positive loading of 0.50 on PC2, while variables in Test 2.2 phase (var15_test2.2, var16_test2.2) show a negative loading of −0.50, forming a cross-phase differential pattern. When the value of PC2 increases, the gait variance in Test 2.1 phase is significantly greater than that in Test 2.2. This is due to individual nervousness or conservative strategies during the initial test, leading to frequent gait adjustments. Conversely, when the value of PC2 decreases, the gait variance in Test 2.2 phase surpasses that in Test 2.1, indicating a decline in control ability caused by physical exhaustion and decreased attention due to repeated testing. Therefore, PC2 features the distinct response patterns of the neuromuscular system during repetitive tasks, either through learning-based optimization strategies or functional decline due to fatigue. The correlation between PC2 and duration is also 0.24, indicating a similar slight positive trend.

These findings not only highlight the role of gait variability in performance but also provide insights into how physiological strategies (such as gait adjustments) and fatigue influence individual test outcomes, contributing to a more comprehensive assessment of performance differences in motor tasks.

### 5.3. Result 2: The Relationship Between Comprehensive Indicators and SPPB Test 2 Scores

The study found that there is a positive correlation between the comprehensive indicators and SPPB Test 2 scores. Wang et al. [7] similarly observed potential links between gait parameters and functional scores in their work with single RGB camera gait analysis. The clinical relevance of these findings, despite not reaching statistical significance, is consistent with observations in other video-based assessment studies, where subtle movement features showed promising but not always statistically significant relationships with clinical measures [6].

Specifically, Test2_1, Test2_2, and Test2 scores showed consistent positive trends with multiple comprehensive indicators, including Mean [24], as shown in Figure 8. For instance, the correlation coefficient between Test2_1 and Mean was 0.29, while the correlation coefficient between Test2_2 and Mean was 0.35, though these values did not reach significance (*p* < 0.05) [21]. Additionally, PC1 and PC2, as two key indicators derived from principal component analysis, also exhibited similar positive trends, further supporting the conclusions. Overall, although these results were not statistically significant, their consistency suggests a potential link between comprehensive indicators and scores, providing a basis for further exploration [39].

## 6. Conclusions

This study utilized a monocular camera and computer vision technology to quantify gait fluctuation features reflected by skeletal keypoints during walking in the elderly, and explored their association with performance on the SPPB [42]. The results revealed a significant negative correlation between the mean fluctuation in walking distance and the duration required to complete the SPPB, suggesting that greater gait variability may indicate better physical coordination and vitality [43]. Furthermore, although some indicators did not reach statistical significance, a composite index constructed via *PCA* showed a positive correlation trend with SPPB subtest scores, implying a potential link between gait fluctuation features and physical functional status [44].

Compared with existing studies, this research presents notable innovations in both methodology and focus. On the one hand, the use of a monocular camera enabled contactless gait data acquisition, eliminating the need for wearable devices commonly required in traditional approaches [42]. This enhances feasibility and operational efficiency, making it particularly suitable for routine screening of older adults in community and home settings [41]. On the other hand, this study examined gait variability from the perspective of temporal fluctuations in walking distance, establishing a more direct relationship with physical functional status than static gait parameters typically used in the literature [44].

In addition, the composite index derived from *PCA* effectively captured individual differences in gait execution, providing theoretical support for personalized fall risk assessment in the elderly [40]. Future research may expand the sample size, incorporate additional gait and physiological parameters, and explore modeling and prediction using deep-learning techniques, thereby improving the accuracy and practical value of functional assessment and fall risk warning in older populations [12,18].

## Figures and Tables

**Figure 1 healthcare-13-01405-f001:**
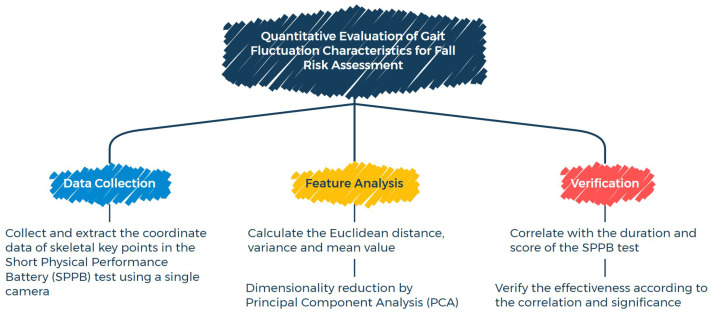
Conceptual diagram.

**Figure 3 healthcare-13-01405-f003:**
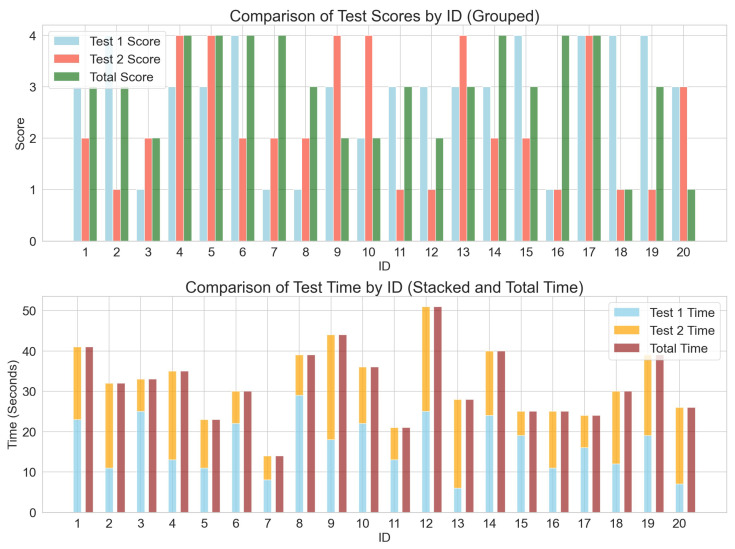
Comparison of Test 2 score and duration.

**Figure 4 healthcare-13-01405-f004:**
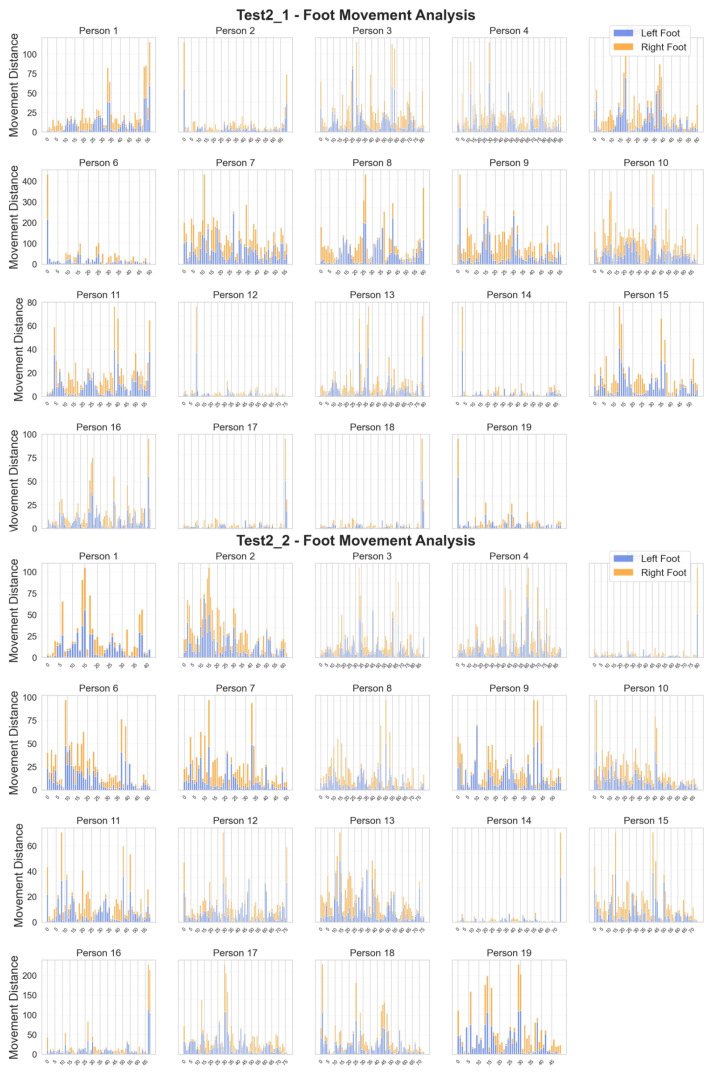
Stacked chart of footstep distance distribution.

**Figure 5 healthcare-13-01405-f005:**
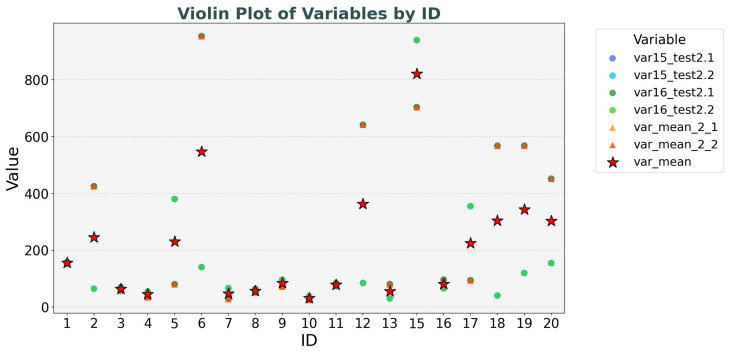
Variance distribution diagram.

**Figure 6 healthcare-13-01405-f006:**
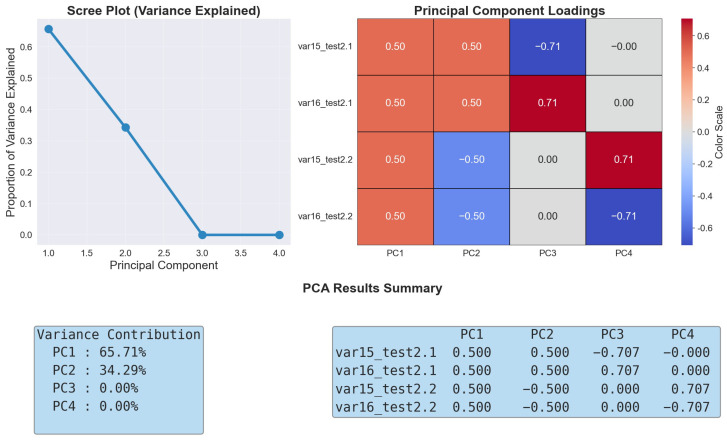
Principal component analysis (PCA) results.

**Figure 7 healthcare-13-01405-f007:**
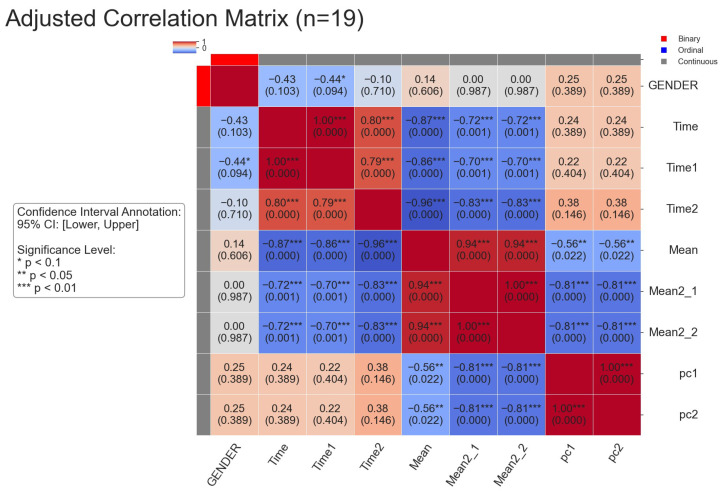
Correlation analysis between Test 2 duration in SPPB and comprehensive indicators.

**Figure 8 healthcare-13-01405-f008:**
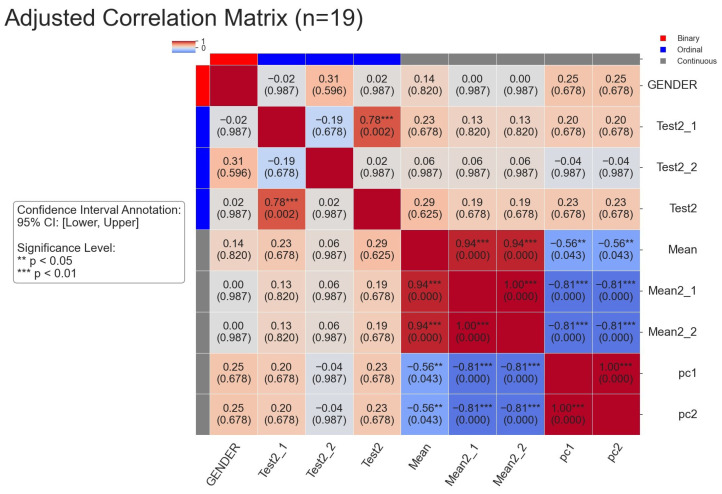
Correlation analysis between Test 2 scores in SPPB and comprehensive indicators.

**Table 1 healthcare-13-01405-t001:** Comparative analysis of research studies: equipment, privacy, and methodologies.

Papers	Equipment Used	Privacy Protection	Camera Capture Method	Quantity of Equipment Used
“Video-Based Activity Recognition for Automated Motor Assessment of Parkinson’s Disease” Sarapata et al. [6]	Consumer-grade handheld devices (e.g., tablets), OpenPose	Privacy is considered. Measures: Obtained informed consent from subjects and followed ethical committee approvals; only used the 2D coordinate data of human joint positions (stick-figure data) extracted by OpenPose, without directly using the original video data.	Static (fixed-camera)	One camera; auxiliary pose estimation tools
“A Single RGB Camera Based Gait Analysis with a Mobile Tele-robot for Healthcare” Wang et al. [7]	Dual robot system, iPad, Intel RealSense D415	Whether privacy is considered and specific measures were not clearly mentioned. It is not stated whether only stick-figure data is used, and the use of original video data may be involved, so privacy protection is questionable.	Dynamic (mobile robot camera)	One RGB-D camera; one iPad; one mobile tele-robot
“Computer Vision System Based on the Analysis of Gait Features for Fall Risk Assessment in Elderly People” Cedeno-Moreno et al. [19]	IDS UI-3130CP-M-GL R2 monochrome camera, treadmill	Privacy is considered. Measures: Approved by the ethics committee and obtained informed consent; used the key-point data (stick-figure data) extracted for gait feature analysis, without directly using the original video data.	Static (treadmill recording)	One monochrome camera; one treadmill
“On the reliability of single-camera markerless systems for overground gait monitoring” Boldo et al. [20]	W Intel RealSense D415, Vicon MX 13 infrared cameras	Privacy is considered. Measures: Obtained ethical approval and informed consent; used the human joint key-point data (stick-figure data) extracted by OpenPose, without directly using the original video data.	Static (overground walking)	One RGB-D camera; eight infrared cameras
“Fall risk prediction using temporal gait features and machine learning approaches” Lim et al. [21]	Two fixed cameras, Mendeley public dataset (inertial sensor data)	Privacy is considered. Measures: Approved by the ethics committee, obtained informed consent, and anonymized the data; used the 26 human key-point data (stick-figure data) extracted by AlphaPose, without directly using the original video data.	Static (TUG test)	Two cameras; public dataset used; no additional hardware
Our study	One fixed camera	Privacy is considered. During data collection, privacy protection is ensured by capturing only the human keypoint data through the camera, without recording actual video footage.	Static (fixed-camera)	One camera

## Data Availability

The original contributions presented in this study are included in the article. Further inquiries can be directed to the corresponding author(s).

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
