# Peer review of "Vision-Based Assessment of Skeletal Muscle Decline: Correlating Gait Variance with SPPB Performance"

_healthcare, 2025, doi:10.3390/healthcare13121405_

Round 1
Reviewer 1 Report
Comments and Suggestions for Authors
This study proposes a non-invasive method to assess fall risks in the elderly by quantifying gait movement characteristics using computer vision. The authors claimed that their method provides a low-cost, contactless approach for screening skeletal muscle decline in community-dwelling elderly, offering a foundation for personalized fall risk interventions. However, there are many areas need to be considered for enhancing the quality of this work.
- The organization and presentation of the article are poor. Authors are suggested to add contents between section and sub-section of the manuscript. Before introducing sub-section please add contents for the section and so on.
- Please provide conceptual diagram of your work in the Introduction section or Methodology section and explain briefly about it.
- Experimental/Simulation environment, diagram for data acquisition approach, data sets & their attributes and data models variation from existing work are not understandable. Tools used for the work are not well presented.
- Please support your results by visible evidences and their justifications.
- Table 1 caption need to be on the top. Please provide description of its interpretation.
- Figure 1 is not readable. Its description is missing. And, it has not been referenced in the document. Similarly, all table, figures and equations are not referenced and well explained.
- Please make the 3rd part of figure 5 more readable.
- I suggest for providing comparative table to show the performance of this work with related work. What about accuracy/precision of the proposed method? Any trade-offs?
- Proof reading, re-formatting, maintaining the flow of writing are essential.
Author Response
Point1: This study proposes a non-invasive method to assess fall risks in the elderly by quantifying gait movement characteristics using computer vision. The authors claimed that their method provides a low-cost, contactless approach for screening skeletal muscle decline in community-dwelling elderly, offering a foundation for personalized fall risk interventions. However, there are many areas need to be considered for enhancing the quality of this work.
Response 1: Thank you for the valuable suggestions you have provided for this research! Your feedback is of utmost importance for improving the quality of the research. I will carefully study each comment, conduct in-depth analysis, and systematically sort out the current deficiencies in the research. In the follow-up, I will optimize and refine the research content with a rigorous attitude, and make every effort to improve the details of the research methods and the presentation of the conclusions to ensure the scientificity and practicality of the research. Thank you again for your professional guidance!
Point2: The organization and presentation of the article are poor. Authors are suggested to add contents between section and sub-section of the manuscript. Before introducing sub-section please add contents for the section and so on.
Response 2: I have taken your suggestions on board and added relevant content between the chapters and sections of the article. Through these adjustments, the transitions in the article have become more natural and smooth, the structure is clearer, and the reading experience is more coherent and understandable. This will help readers better understand the research ideas and content.
Point3: Please provide conceptual diagram of your work in the Introduction section or Methodology section and explain briefly about it.
Response 3: The conceptual diagram of our work has been added, and corresponding descriptions have been included in the text.
Point4: Experimental/Simulation environment, diagram for data acquisition approach, data sets & their attributes and data models variation from existing work are not understandable. Tools used for the work are not well presented.
Response 4: In terms of the experimental/simulation environment and data collection, this study adopted camera capture technology to synchronously record video data during the process when the subjects were conducting the SPPB test. Subsequently, these video data were converted into datasets available for analysis. It should be noted that this study only used the data of 19 subjects for analysis, and the datasets were stored in JSON format.
Regarding the research tools and system, this system combines Google's PoseNet model with TensorFlow.js technology. Using a USB camera as the image input device, it can perform real-time human pose recognition in the browser and output the coordinate results of 17 body parts and their corresponding scores. Meanwhile, with the aid of Raspberry Pi 3B and in conjunction with tools such as OpenCV and Node-Canvas, data collection is carried out at one-second intervals, and SparkStreaming technology is utilized for data processing. By analyzing the skeletal region information output by the PoseNet model, this system achieves the functions of assessing the quality of in-home physical activities and location sensing. In addition, a Web visualization system has been developed to process the collected data in the cloud and display it in the form of charts, ultimately realizing fine-grained and continuous physical activity monitoring as well as simple in-home positioning functions.
Point5: Please support your results by visible evidences and their justifications.
Response 5: I have made revisions to the results section to make the research results clearer and more reasonable.
Point6: Table 1 caption need to be on the top. Please provide description of its interpretation.
Response 6: I have moved the caption of Table 1 to the top as required. Additionally, I have provided a detailed description of the table's content to facilitate better understanding.
Point7: Figure 1 is not readable. Its description is missing. And, it has not been referenced in the document. Similarly, all table, figures and equations are not referenced and well explained.
Response 7: In Figure 1, the font size has been adjusted to make it easier to read. At the same time, citations have been added to the thesis. For some of the figures, since the citations added in the title position may not be obvious, the positions of the citations will be readjusted to make them more noticeable. Additionally, explanations have been added for the formulas that were not previously mentioned.
Point8: Please make the 3rd part of figure 5 more readable.
Response 8: Figure 5 has been adjusted to be more readable.
Point9: I suggest for providing comparative table to show the performance of this work with related work. What about accuracy/precision of the proposed method? Any trade-offs?
Response 9: Thank you for your valuable suggestions! I fully agree with the importance of presenting the performance advantages through a comparison table. However, due to the limited length of the article, adding a new table may affect the overall structure. I sincerely hope that you can provide guidance: Is it possible to delete or reduce some existing content to accommodate the table? At the same time, I also hope that you can give suggestions on the core comparison indicators and design format of the table, so as to optimize the paper better. Thank you for your guidance!
Point10: Proof reading, re-formatting, maintaining the flow of writing are essential.
Response 10: hank you very much for the valuable comments provided by the reviewers! These suggestions have pointed out the key directions for the optimization of the paper and are of great significance for improving the quality of the paper. I have made comprehensive and substantial revisions to the clarity of the statistical presentation, the writing style, and the demonstration of novelty in the paper. I strive to present a more rigorous argumentation, a smoother expression, and a more comprehensive interpretation of innovation, so that the paper can meet the requirements for publication. Thank you again for your professional guidance!

Reviewer 2 Report
Comments and Suggestions for Authors
The topic is timely and highly relevant given the growing elderly population and the need for scalable, low-cost, non-invasive screening tools. The paper is generally well-organized, with a clear motivation and structure. However, while the study shows potential, there are several areas requiring clarification, methodological refinement, and improvement in the writing style.
The use of a single RGB camera for gait analysis and the emphasis on spatiotemporal foot movement variance are noteworthy. The proposal of a composite index derived from PCA is a good attempt to synthesize gait variability. However, the novelty is somewhat diluted by the lack of a clear distinction from prior studies using vision-based systems for gait assessment. The manuscript needs to explicitly clarify what differentiates this method from previous works like Wang et al. [6] or Sarapata et al. [5].
The methodology is described in reasonable detail, but the sample size (N=19) is too small to draw generalizable conclusions, particularly for correlation-based findings. Also, no power analysis is mentioned. PCA is used appropriately, but the rationale for reducing to exactly two components should be better justified statistically. The explanation of how the composite indicator was constructed could be clearer, particularly in how it integrates variance measures from both feet and both tests.
Recommendations: Include a justification for the sample size or discuss its limitations; Expand the explanation of the indicator formulation (e.g., how Eq. (3) was defined and validated); Consider including confidence intervals or p-values for correlation coefficients.
The use of multiple correlation coefficients (Pearson, Spearman, Kendall's Tau) is methodologically sound, but it is unclear how decisions were made regarding which metric to use for which variable pair. The statistical significance is reported only for some correlations. Some relationships are described as "positive trends" without significance, which weakens the conclusions.
Recommendations: Include a table summarizing all correlation values, significance levels, and sample sizes; Be more cautious in interpreting non-significant trends, especially in such a small sample.
This article addresses an important problem and has the potential to contribute to fall risk assessment methods. However, substantial revisions are needed in terms of statistical clarity, writing style, and justification of novelty before it can be considered for publication.
Comments on the Quality of English Language
The English writing is mostly clear but requires some improvements.
Avoid expressions like “this study plans to...” or “we will...” in a completed study. Use past tense consistently.
Examples: Change "This study plans to use..." to “This study used..."; Instead of "Fig 3) shows the distribution diagram", use "Figure 3 shows the distribution of...".
Author Response
Points1:The topic is timely and highly relevant given the growing elderly population and the need for scalable, low-cost, non-invasive screening tools. The paper is generally well-organized, with a clear motivation and structure. However, while the study shows potential, there are several areas requiring clarification, methodological refinement, and improvement in the writing style.
Response 1: Thank you very much for your meticulous and professional feedback! Your comments are of great value to us in further improving our research, enabling us to clearly recognize the current deficiencies and areas for improvement in our paper. Regarding the content that you mentioned needs to be clarified, we will immediately sort out the relevant logic and supplement the necessary explanations to ensure rigorous argumentation and reliable conclusions. As for the optimization of the methodology, I will make revisions according to the suggestions you provided. In terms of writing style, we will also carefully polish the sentences to enhance the fluency and readability of the text. After the subsequent revisions are completed, we sincerely hope to receive your guidance and suggestions again.
Point2:The use of a single RGB camera for gait analysis and the emphasis on spatiotemporal foot movement variance are noteworthy. The proposal of a composite index derived from PCA is a good attempt to synthesize gait variability. However, the novelty is somewhat diluted by the lack of a clear distinction from prior studies using vision-based systems for gait assessment. The manuscript needs to explicitly clarify what differentiates this method from previous works like Wang et al. [6] or Sarapata et al. [5].
Response 2: Thank you, reviewers, for pointing out the issue! There are significant differences in research indicators between this study and the studies by Wang et al. [6] and Sarapata et al. [5]. Wang et al. [6] and Sarapata et al. [5] mainly focused on conventional indicators such as step length, step frequency, and time during walking, but they did not conduct in-depth research on the volatility of gait movement. In contrast, this study focuses on analyzing the volatility and mean value of gait. By deriving comprehensive indicators through Principal Component Analysis (PCA) to quantify gait variability, it provides a new perspective and method for in-depth analysis of the spatio-temporal variation patterns of gait. In comparison, this study has achieved breakthroughs and innovations in the selection of indicators and the depth of analysis, further highlighting the unique value and significance of the research.
Point3:The methodology is described in reasonable detail, but the sample size (N=19) is too small to draw generalizable conclusions, particularly for correlation-based findings. Also, no power analysis is mentioned. PCA is used appropriately, but the rationale for reducing to exactly two components should be better justified statistically. The explanation of how the composite indicator was constructed could be clearer, particularly in how it integrates variance measures from both feet and both tests.
Response 3: Thank you so much for your valuable suggestions. We are fully aware of the limitation of the small sample size (N = 19) in our current study. During the research process, we have already realized that this may affect the generalizability of our research findings, especially considering that our study is correlation-based. In the results section, we will be more cautious when interpreting and analyzing the results, clearly emphasizing the potential limitations brought about by the limited sample size. In addition, we recognize the importance of conducting a power analysis. Although the power analysis is not included in the current manuscript, we will consider incorporating relevant discussions on the power analysis in future revisions to enhance the comprehensiveness and reliability of our research.
In the chapter "The relationship between comprehensive indicators and Test 2 duration in SPPB", we have added an in - depth explanation of the statistical basis for applying Principal Component Analysis (PCA).
In the chapter "Calculate the Variance of Movement Distance and Construct a Comprehensive Indicator", the calculation details of the variance measurement values of both feet and the two tests have been added, and the relevant formula explanations have been supplemented. At the same time, the reasons for constructing the comprehensive indicator have been expounded.
Point4:Recommendations: Include a justification for the sample size or discuss its limitations; Expand the explanation of the indicator formulation (e.g., how Eq. (3) was defined and validated); Consider including confidence intervals or p-values for correlation coefficients.
Response 4: I have comprehensively improved the thesis. In the "Purpose and Overview" section, the rationality basis and limitation analysis of the sample size selection have been supplemented, making the research foundation more persuasive. In the sections of "Step 1: Pedestrian Movement Data Extraction and Analysis" and "Step 2: Calculate the Variance of Movement Distance and Construct a Comprehensive Indicator", the definition and interpretation as well as the verification process of Formula (3), and the detailed description of the construction process of the comprehensive indicator have been added respectively, enhancing the logicality and transparency of the indicator construction. At the same time, Figures 7 and 8 have been revised, and the confidence intervals or p-values of the correlation coefficients have been supplemented, further improving the reliability and rigor of the research results.
point5:The use of multiple correlation coefficients (Pearson, Spearman, Kendall's Tau) is methodologically sound, but it is unclear how decisions were made regarding which metric to use for which variable pair. The statistical significance is reported only for some correlations. Some relationships are described as "positive trends" without significance, which weakens the conclusions.
Response 5: I have made targeted revisions to the paper. In the section "Result 1: The relationship between comprehensive indicators and Test 2 duration in SPPB", the decision-making basis for selecting specific measurement criteria for each variable pair when using different correlation coefficients (Pearson, Spearman, Kendall's τ) has been newly added, clearly expounding the selection logic. At the same time, the content described as "positive trend" in the paper has been revised, and relevant statistical significance information has been supplemented to avoid vague expressions, thereby enhancing the rigor and persuasiveness of the research conclusions.
Point6:Recommendations: Include a table summarizing all correlation values, significance levels, and sample sizes; Be more cautious in interpreting non-significant trends, especially in such a small sample.
Response 6: According to the suggestions, I have clearly marked the correlation coefficients, significance levels, and sample sizes in the chart. Additionally, in the conclusion section, I have made more rigorous modifications to the interpretation of the non-significant trends in the case of small sample sizes.
Point7:This article addresses an important problem and has the potential to contribute to fall risk assessment methods. However, substantial revisions are needed in terms of statistical clarity, writing style, and justification of novelty before it can be considered for publication.
Response 7: Thank you very much for the valuable comments provided by the reviewers! These suggestions have pointed out the key directions for the optimization of the paper and are of great significance for improving the quality of the paper. I have made comprehensive and substantial revisions to the clarity of the statistical presentation, the writing style, and the demonstration of novelty in the paper. I strive to present a more rigorous argumentation, a smoother expression, and a more comprehensive interpretation of innovation, so that the paper can meet the requirements for publication. Thank you again for your professional guidance!

Reviewer 3 Report
Comments and Suggestions for Authors
Some points need improvements. Please see my comments below…
INTRODUCTION
P1L26 - “Traditional methods of assessing fall risks in the elderly often rely on clinical tests such as the Short Physical Performance Battery (SPPB), which, while effective...” – Reference that define effectiveness of this tool on fall risk assessment is needed… The relationship between this tool and falls must be unequivocal…
P2L38 - “In this study, a single camera is utilized to capture the skeletal key point data of the subjects, with a focus on analyzing the spatiotemporal fluctuation characteristics of the movement distances of both feet.” – This gait parameter was analysed based on what literature?… Is this parameter related to falls? Why did you analysed this parameter? This must be clarified… I believe that the following reference could be important: Silva, J., Atalaia, T., Abrantes, J. & Aleixo, P. (2024). Gait biomechanical parameters related to falls in the elderly: a systematic review. Biomechanics 4(1), 165-218, https://doi.org/10.3390/biomechanics4010011
P2L45 - “It’s worth noting that this method using a single-camera setup offers a low-cost and contactless solution for screening skeletal muscle decline in the elderly…” – I believe this method allows to evaluate gait instead skeletal muscle decline…
The study aim is not clear…
METHODS
Exclusion and inclusion criteria of participants?
The description of the gait assessment protocol is needed… about this, nothing is described…
Author Response
Some points need improvements. Please see my comments below…
INTRODUCTION
Point1:P1L26 - “Traditional methods of assessing fall risks in the elderly often rely on clinical tests such as the Short Physical Performance Battery (SPPB), which, while effective...” – Reference that define effectiveness of this tool on fall risk assessment is needed… The relationship between this tool and falls must be unequivocal…
Response 1: When responding to the question of "needing to define the reference for the effectiveness of the Short Physical Performance Battery (SPPB) in fall risk assessment and clarifying the relationship between this tool and falls", I found the research findings of Lauretani, F., and others through consulting relevant literature. This research clearly shows that the total score of the SPPB and the balance ability are directly related to fall incidents, and its assessment efficacy is comparable to that of the commonly used POMA tool. Based on this, I cited this research conclusion in my response, supplementing the explanation of the effectiveness of the SPPB in fall risk assessment, thus clarifying the relationship between this tool and falls. At the same time, I also mentioned the limitations of such traditional tests, which are time-consuming and require professional personnel to administer, making the response more complete and comprehensive.
Point2:P2L38 - “In this study, a single camera is utilized to capture the skeletal key point data of the subjects, with a focus on analyzing the spatiotemporal fluctuation characteristics of the movement distances of both feet.” – This gait parameter was analysed based on what literature?… Is this parameter related to falls? Why did you analysed this parameter? This must be clarified… I believe that the following reference could be important: Silva, J., Atalaia, T., Abrantes, J. & Aleixo, P. (2024). Gait biomechanical parameters related to falls in the elderly: a systematic review. Biomechanics 4(1), 165-218, https://doi.org/10.3390/biomechanics4010011
Response 2:I'm extremely grateful for the recommendation of the literature by Silva et al. (2024), which proves to be of immense value to my research. Referencing the systematic review in this study, it was found that common gait biomechanical parameters such as walking speed and step length are closely associated with falls in the elderly. This provides a strong literature basis for my analysis of the spatiotemporal fluctuation characteristics of bipedal movement distance. By examining both distance and fluctuation dimensions, this study can more comprehensively reflect an individual's gait pattern, which is closely linked to fall risk. As there is currently limited systematic research on this parameter in academia, analyzing it in this study aims to deeply explore the relationship between gait patterns and falls, and provide new perspectives and potential indicators for fall risk assessment, thus filling a research gap.
Point3:P2L45 - “It’s worth noting that this method using a single-camera setup offers a low-cost and contactless solution for screening skeletal muscle decline in the elderly…” – I believe this method allows to evaluate gait instead skeletal muscle decline…
Response 3:Thank you for your correction! We have revised the corresponding content according to your suggestion. In the future, we will check the article more carefully to avoid similar ambiguities and ensure that the expressions are accurate and clear.
Point4:The study aim is not clear…
Response 4:Thank you for pointing out this issue. I've noticed that during the research process, the main focus was on analyzing the relationship between gait characteristics and physical indicators. However, the original study aim regarding finding a long - term and effective detection method was not clearly presented, and the objective related to skeletal muscle health was ambiguous. In response to this, I have revised the research objectives section of the article to make the aims more explicit, coherent, and in line with the actual research content. This ensures that the research goals are clearly communicated to the readers, enhancing the clarity and integrity of the paper.
METHODS
Point5:Exclusion and inclusion criteria of participants?
Response 5:Thank you for pointing out this issue! The data of the subjects in this study is sourced from a specific data collection channel that has been previously communicated and explained to your journal. Since the data mainly comes from the individuals who participated in the tests at the hospital, the specific inclusion and exclusion criteria are not clearly marked in the original information. The basic information of the subjects presented in the article currently (including age, basic physical condition, etc.) has fully reflected the characteristics of the population of this batch of data. If it is necessary to further supplement the relevant criteria, we will actively communicate with the data provider and do our best to improve the relevant content.
Point6:The description of the gait ssessment protocol is needed… about this, nothing is described…
Response 6: Thank you for your valuable feedback. I have added supplementary descriptions of the gait assessment in the "Proposed Method" section to address this issue.

Round 2
Reviewer 1 Report
Comments and Suggestions for Authors
Most of the comments have been adjusted and the article has been updated accordingly. Therefore, now the manuscript is ready for further processing from my side.
Reviewer 2 Report
Comments and Suggestions for Authors
This version represents a clear and significant improvement over the first version. It refines the scientific communication, expands methodological and analytical depth, and strengthens the practical relevance and theoretical contributions of the study.
Reviewer 3 Report
Comments and Suggestions for Authors
None.